# Occurrence and Distribution of Uranium in a Hydrological Cycle around a Uranium Mill Tailings Pond, Southern China

**DOI:** 10.3390/ijerph17030773

**Published:** 2020-01-26

**Authors:** Wenjie Ma, Bai Gao, Yadan Guo, Zhanxue Sun, Yanhong Zhang, Gongxin Chen, Xiaojie Zhu, Chunyan Zhang

**Affiliations:** Fundamental Science on Radioactive Geology and Exploration Technology Laboratory, East China University of Technology, NanChang 330013, China; wenjiema@ecut.edu.cn (W.M.); guoyadan@ecut.edu.cn (Y.G.); zhxsun@ecut.edu.cn (Z.S.); yanhongzhang@ecut.edu.cn (Y.Z.); gxchen@ecut.edu.cn (G.C.); miloomoecit@163.com (X.Z.); m13047903981@163.com (C.Z.)

**Keywords:** water–rock interactions, critical zones, uranium contamination, Aquifer, hazard assessment

## Abstract

Uranium (U) mining activities, which lead to contamination in soils and waters (i.e., leachate from U mill tailings), cause serious environmental problems. However, limited research works have been conducted on U pollution associated with a whole soil-water system. In this study, a total of 110 samples including 96 solid and 14 water samples were collected to investigate the characteristics of U distribution in a natural soil-water system near a U mining tailings pond. Results showed that U concentrations ranged from 0.09 ± 0.02 mg/kg to 2.56 × 10^4^± 23 mg/kg in solid samples, and varied greatly in different locations. For tailings sand samples, the highest U concentration (2.56× 10^4^ ± 23 mg/kg) occurred at the depth of 80 cm underground, whereas, for paddy soil samples, the highest U concentration (5.22 ± 0.04 mg/kg) was found at surface layers. Geo-accumulation index and potential ecological hazard index were calculated to assess the hazard of U in the soils. The calculation results showed that half of the soil sampling sites were moderately polluted. For groundwater samples, U concentrations ranged from 0.55 ± 0.04 mg/L to 3.36 ± 0.02 mg/L with a mean value of 2.36 ± 0.36 mg/L, which was significantly lower than that of percolating waters (ranging from 4.56 ± 0.02 mg/L to 12.05 ± 0.04 mg/L, mean 7.91 ± 0.98 mg/L). The results of this study suggest that the distribution of U concentrations in a soil-water system was closely associated with hydrological cycles and U concentrations decreased with circulation path.

## 1. Introduction

Uranium (U) is a strategic source and has been enormously exploited with an increased demand of nuclear power industry. Generally, there are three ways to exploit U: surface (open pit), underground and solution mining (e.g., in-situ leaching). Tailings would be generated worldwide during the U mining and ore processing with particular means of open pit and underground [1]. Presently, the world’s stockpile of U waste rocks was estimated to exceed 40 billion tons, and tailings 20 billion tons [2,3]. With an advent of U-free markets, many mines became non-economically effective and were finally shut down. However, tailings and waste rocks produced were difficult to be efficiently isolated from the environment. They can be a significant source of radionuclides being released in soils and natural waters via weathering, suspension, denudation, percolation and erosion of meteoric water. These processes would further expand pollution and cause a great damage to the ecological environment [4,5,6].

The impact of radionuclides produced during the processes of U mining has significantly received public concern. The chemical and radiological toxicity of U shows an obvious biological effect and poses a threat to humans [7]. U occurs widely in soils, natural waters, air, plants, and animals [8,9,10]. Long-term intake of U-contaminated groundwater or crops grown in contaminated soils causes a great damage to human health. Cases have been reported that genotoxic effects and immunotoxicity resulting from U. Analyses of trace elements and C reactive protein were carried out for blood samples collected from residents living near a U mine to investigate the impact of U mining, showing that it did cause some public health problems to the residents [11]. Thus, it is necessary to investigate the characteristics of the pollutants in soil-water systems near a U mill tailings ponds.

A lot of studies have been carried out to investigate the characteristics, and to assess the environmental impact of U [12,13,14,15,16]. Wu et al. studied the concentrations of U in natural waters at Datong Basin and found that 24% of the groundwater samples had U concentration above the WHO guideline and the average U concentration for surface water was 5.80 mg/L. Mishra et al. studied the distribution of U concentrations in soils affected by a nuclear power plant accident, concluding that the mobility of U was highly dependent on soil characteristics in the particular area. However, their works were relatively simple without fully considering the relationship of U distribution in each part of a hydrological cycle, which is essential to understand the migration and transport of U in the environment. Therefore, a representative study area near the storage site of radioactive waste was selected to study the occurrence, distribution and behavior of U in a whole hydrological cycle (the conceptual model shown in Figure 1), and the specific objectives are to: (i) compare U concentrations in different geomatrices (i.e., soils, surface and ground water); (ii) characterize the impact factors controlling U concentrations in soil and water systems; (iii) assess the tendency of U migration during the hydrological cycle.

## 2. Materials and Methods

### 2.1. Study Area

The U mine was the largest volcanic type ores deposited in China with an area of 360 km^2^. The mining activities have lasted for over 50 years and it was shut down just in recent years. The U ore was mined by hydrometallurgy and then was neutralized with lime to form tailings which were stored in tailings pond. As a result, more than 500,000 tons of waste rocks, and more than 2 million tons of tailings have been produced. The tailings pond was bounded by mountains on three sides, belonging to the valley type. The catchment area of the tailings pond was 1.63 km^2^, and the dam of the tailings pond was initially used in 1973.

The study area was located in subtropical climate zone, with high temperature, heavy rain and strong weathering in summer. Geomorphologically, the catchment area was mainly high in the center and low in the periphery, and the mountain ranges were located in the northeast. The average annual temperature was 17.6 ℃, and the average annual rainfall was 1774 mm. The main crop around the tailing pond was rice and groundwater were mainly hosted by bedrock fissures which were recharged by precipitation and discharged to surface rivers.

### 2.2. Sample Collection and Analysis

#### 2.2.1. Sampling

To investigate U concentrations and distribution in various media of a hydrological cycle, four types of samples including tailings sand, paddy soil, percolating water and groundwater were collected. For tailings sand, four sampling points were selected and were labeled with sequential codes (T1, T2, T3 and T4) (See Figure 1). At each sampling point, samples were taken from six layers with depths at 0, 20, 40, 60, 80 and 100 cm below land surface, respectively. About 1 kg of each sandy sample was collected in a cloth bag and was sealed immediately for preservation.

Similarly, for paddy soil, samples were collected from 12 sampling points at the depth of 0, 20, 40, 60, 80 and 100 cm below land surface, respectively. To avoid cross-contamination, sampling location were distributed in cross within a grid of 100 m × 100 m (Figure 2). After about 1 kg soil being collected, it was preserved in a cloth bag with label from S1 to S13, as shown in Figure 2.

Seven percolating water samples (P1 to P7) and seven groundwater samples (G1 to G7) were collected around the tailings pond and/or from local villages. All samples were filtered by 0.45 μm-pore-sized filters, and were stored in 0.5 L sampling vessels with addition of 65% HNO_3_ to pH < 2.0. The location of sampling points was shown in Figure 2.

#### 2.2.2. Analysis

Parameters including T, pH and Eh were determined using multiple-parameter (HACH HQ40d, USA) meter in-situ. Concentrations of U were analyzed by using ICP-OES (Agilent 5100VDV, USA) with a detection limit of 0.01 mg/L. Solid samples (tailings sand and paddy soil) were analyzed after full digestion. Briefly, the tailings sand samples were dried in a drying oven at 105–110 ℃, and then were ground to < 200 mesh. A certain amount of sample was fed in a centrifuge tube where sulfuric acid (4%) was added for 30 hours digestion. The mixture was then centrifuged and 10 mL supernatant was obtained for measurement. The paddy soil samples were dried before being crushed. After removed the residual roots and other sundries, the ground sample was digested by hydrofluoric acid (density: 1.13× 10^6^ mg/L) and nitric acid (density: 1.42× 10^6^ mg/L) for final analyses.

During the course of determination, six standard solutions and one blank solution were running to check the stability of the system for U measurements. Each standard solution was tested three times, and the mean measured values were used to establish standard curve, the relative error was better than 1%. U concentrations were also tested three times for each sample. Finally, the liquid concentration (mg/L) was transferred to solid concentration (mg/kg) for tailings sand and paddy soil samples.

Error analysis was carried out to deal with the three times repeated measurements and the mean value of U concentration. The standard error of three test results was better than 5% for all samples, and the mean value with standard error was used as the result of each sample for subsequent discussions. Pearson correlation analysis was done by using IBM SPSS Statistics 20.0 program with a confidence level of 99 % to determine the correlation between average concentrations of U and organic matter at sampling depth. Additionally, a method of K-S inspection was employed for each type sampling average U concentration (including tailings sand, paddy soil, percolating water and groundwater) to check the reliability of sample collection. Values obtained from the inspection analysis followed the normal distribution at a 0.05 confidence interval.

## 3. Results and Discussion

### 3.1. Tailings Sand Samples

Total concentrations of U in each tailings sand samples were shown in Figure 3. U concentrations ranged from 1.76 × 10^4^ ± 17 mg/kg to 2.56 × 10^4^ ± 23 mg/kg, and showed vertical variation with all highest values at the depth of 80 cm. The average value of U concentration for all four sand samples at 80 cm was 2.49× 10^4^ ± 2.40× 10^2^ mg/kg, while U concentrations for other five layers were approximately 2.00 × 10^4^ mg/kg. There was no obvious difference in the total U concentrations among different sampling points (T1: 1.23 × 10^5^ mg/kg, T2: 1.24 × 10^5^ mg/kg, T3: 1.24 × 10^5^ mg/kg and T4: 1.24 × 10^5^ mg/kg). It indicates that U was distributed evenly in the tailings sand and migrated consistently in the horizontal direction.

There were two main reasons for U distribution characteristics. Firstly, precipitation played a role in driving U leaching from sandy solids. This tailings pond was located in a subtropical climate area with high temperature and much rain, especiallyE acid rain, although the tailing pond was alkaline with a pH range of 7–10. After being leached by acidic meteoric water, U-bearing rocks dissolved and solution pH became alkaline. Accordingly, U concentrations in the pore water were increased. Therefore, the annual and seasonal rainfall affected the moisture content [17], redox potentials and pH values in tailings pile. These parameters influenced U dissolution, migration and precipitation via water-rock interaction along the sampling profile. Another process controlling U behavior was that rainfall percolation moving downwards through the sand profile depending on the hydraulic resistance both in the top and bottom layers. This was largely due to the particle-size distribution, the gravel and sand being highly permeable. In the tailings pond, the particle size was about 0.42 mm with a percent of 85%. With long-term weathering, coarse sand and fine mud can be easily isolated, which generated a smaller density and worse permeability in tailings area, and then affected the distribution of U in the profile. Furthermore, with an increase in depth, the compaction of the lower layer enhanced, which also influenced the concentrations of U in the sandy samples.

### 3.2. Percolating Water Samples

Due to the climate condition in study area, the tailings were under a non-steady state. The moisture derived from acid infiltration allowed a long-term seepage to migrate from the disposal cell. The discharge water posed pollution hazards to the surrounding environment, including soils, surface water and groundwater. This has been a major environmental concern in U mill tailings [18]. We collected seven samples including four discharge water and three leakage water around the U tailings pond (Table 1).

Concentrations of U in percolating water varied from 4.56 ± 0.02 mg/L to 12.05 ± 0.04 mg/L with an average value of 7.91 ± 0.98 mg/L. The maximum U concentration was found in tailings pond discharge water, which may be attributed to the rainy season when the surface water levels raised, and flooding derived U migration from the pond to the natural water body. The pH of the seven samples ranged from 4.52 to 8.66 with an average of 6.26. The lowest pH value (4.52) occurred in the farmland water sample near the bottom of tailings pond dam. This was because that the pH value at the bottom of the dam was low, which affected the pH values of the surrounding water, and agricultural activities may also have an impact on pH values of the surrounding water.

According to the regulations of radiation protection and environmental protection of U mining and metallurgy (GB23727-2009), the effluent U concentration of waste water was limited to 0.3 mg/L in discharge outlet, and 0.05 mg/L was limited for the first point of water intake. However, U concentrations in the discharge or leakage water samples were all far above these standard limits, particularly the pond discharge water whose U concentration was 40 times higher than the threshold of regulation of radiation and environmental protection. U concentrations of leakage water at the top of tailings pond dam were much higher than those at the bottom, which was consistent with the analytical results of tailings sands samples.

Another important characteristic for percolating water was that U concentrations in the discharged water decreased substantially after the sewage treatment plant, but were still relatively high in the drain water of surrounding farmland and residential villages. It indicated that the drain water came from other water resources like leakage water rather than the water from treatment plant.

### 3.3. Paddy Soil Samples

Extensive agriculture activities were generally found in the study area mainly irrigated by surface and ground waters nearby. Soils, as the tail end of the hydraulic circulation route, tend to accumulate a large amount of pollutants. Studies have shown that high U concentrations occurred in the soils and vegetation near the U mine, which impose a threat to the local ecological environmental [19,20,21,22]. The concentrations of U in each soil section were shown in Figure 4.

U concentrations in soil samples ranged from 0.09 ± 0.02 mg/kg to 5.22 ± 0.04 mg/kg with an average value of 1.96 ± 0.15 mg/kg. 17% samples were higher than the background value of China with 2.80 mg/kg and 9% were larger than the background value of Jiangxi province with 4.40 mg/kg [23]. U accumulating in the surface layer was 2–6 times higher than in the bottom, which was different from the distribution feature of U in the tailing sand profile. The relatively high U concentrations in top soil may be related to phosphate fertilization, owing to Ca-U-P precipitation [24]. In addition, due to the development of root system and strong respiration in the surface layer, the oxygen showed low concentrations in soil, leading to a reduction condition, where U was not easy to dissolve and migrate. Moreover, the free U ions in the rhizosphere of plants may form a stable U-chelate complex with some chelating agents being secreted by roots, forcing U to accumulate around the roots.

There were differences in distribution of U in transversal and longitudinal directions. In transverse direction, U concentrations roughly decreased from sampling site 13 to site 9. This was consistent with the change of elevation. While in longitudinal direction, the farther from the tailings pond, the lower concentrations of U, which was consistent with the previous study [25]. There were three main factors to affect the vertical changes of U concentrations. The first one was the natural attenuation processes [26]. The second was the pH values, since average pH value was 4.2 in the surface layer and turned to 6.43 at the bottom where the migration of U was hindered. The third one was the presence of organic matters [27,28]. Organics can adsorb U effectively, although desorption may occur due to competition between mineral surface sites and dissolved ligands [29]. The average content of organic matter in the soil samples was 7.70 × 10^3^ mg/kg, 1.03 × 10^4^ mg/kg, 1.64× 10^4^ mg/kg, 2.37× 10^4^ mg/kg, 3.24× 10^4^ mg/kg, 4.33× 10^4^ mg/kg at each layer from 0 to 100 cm below the surface, respectively, and showed a good positive correlation with the average U concentration with a correlation coefficient of 0.93.

Although U concentrations in the soils were much lower than those in tailings samples, it could still be detected at 100 cm below the surface. Therefore, it was necessary to evaluate whether the soil had been contaminated. Geo-accumulation index, which not only reflected the natural variation characteristics of U [30], but also identified the impact of human activities on the environment, was used. Potential ecological hazard index which considered the action of toxicity characteristics was also used to assess the ecological impact of the soils. The value 3.21 mg/kg of the topsoil sample on the hill side next to the tailings pond was selected as the regional background value to evaluate from a more representative viewpoint. The results showed that, the I_geo_ ranged between 0 and 1 for site 1, 2, 3, 4, 6 and 7 samples, showing moderate levels of pollution. The results of potential ecological hazard assessment showed that half of the sites were under medium ecological hazard.

### 3.4. Groundwater Samples

Due to increasing consumption of groundwater and enhanced reliability of groundwater as the primary source of water supplies, the quality of groundwater has attracted much attention. Groundwater was buried at depth of 0-5 m in the study area. The aquifer systems were mainly replenished by atmospheric precipitation and surface water, while was discharged via springs in low-lying terrain. Abundant precipitation renders water circulations more frequent between surface water and groundwater, which provided a convenient natural condition for U migration from the tailings pond into aquifers. Groundwater was used for irrigation and drinking in this area. Seven representative groundwater samples were taken from the villages around the tailings ponds (Table 2).

Dissolved U concentrations in groundwater ranged from 0.55 ± 0.04 mg/L to 3.36 ± 0.02 mg/L with an average value of 2.36 ± 0.36 mg/L, being all above the threshold of groundwater standards for tailings and inactive sites of USEPA, and also were higher than the regulation of radiation and environmental protection in U mining of China (GB23727-2009). The highest U concentration being found in G2 was 67 times higher than the threshold set by GB23727-2009, and was up to 76 times higher than the threshold set by USEPA. Due to the infiltration of oxygen-rich groundwater, partial oxygen pressure of the groundwater increased, leading to an oxidative condition [31], which favored dissolutions of U-bearing minerals contained in historical accumulated tailings sand in aquifers. It was also found that, in sampling site G2 and G3, which were located near the river, U concentrations were much higher as compared to those in the site G1, which was far away from the river. This may be due to hydraulic connections between groundwater and surface water, which provided favorable conditions for U migration. Accordingly, it was possible that the difference in U concentrations at different sampling sites may be related to the distance from the river. A similar case has been reported by Anita Eross et al. [32], showing that the occurrence of radionuclides in groundwater was strongly affected by hydraulic regimes.

The percolating water could be a source of U in groundwater. In order to investigate the relationship of U concentration between surface water and groundwater, the variations of pH value as a function of U concentrations in both percolating water and groundwater were shown in Figure 5 as a comparison.

It was obviously observed that the U concentrations in percolating water were much higher than those of groundwater. An increase of U concentration with pH increasing from about 6 to 9 was observed, although three percolation water samples had relatively high U concentration with lower pH values. Aqueous pH value greatly influences U mobilization in natural waters [33]. The main reason was that occurrence of U in groundwater was related to the composition of the soils and rocks, which could act as source of U in waters via dissolution and/or chemical weathering processes that are strongly dependent on pH value. Under the regulation of solution pH, anion exchange, complexation and adsorption were previously suggested to decrease U concentrations in groundwater [13,34,35].

## 4. Conclusions

The occurrence and distribution of U in a water-soil system near a shut-down U mine was investigated. Results showed that U migration and accumulation varied substantially in both water and soil, and was greatly influenced by redox conditions and geological settings. U concentrations ranged from 0.09 ± 0.02 mg/kg to 2.56 × 10^4^ ± 23 mg/kg in tailing sand samples and were much higher than those of paddy soil samples (0.09 ± 0.02 mg/kg to 5.22 ± 0.04 mg/kg). Due to filtration and changing acid-base conditions, the highest U concentration was found at 80 cm below land surface in the sand profile. However, U was largely accumulated at surface layers in the soil profile, which was probably attributed to the influence of organic matters. Percolating water (from mining site) had U concentrations ranging from 4.56 ± 0.02 mg/L to 12.05 ± 0.04 mg/L, being higher than those in groundwater (0.55 ± 0.04 mg/L to 3.36 ± 0.02 mg/L), which showed a wider range of pH values. Thus, after being mobilized from solids, U was dissolved and transported with water infiltration and flow down-gradient. Mineral dissolution, adsorption and complexation reactions modified U concentrations vertically and along a water flow path. The processes for a large part were controlled by solution chemistry and pH values, as suggested by an increase of aqueous U concentrations with sampling location approaching the tailing ponds. Therefore, the geochemical behaviors of U around a U mill tailings pond were regulated by the interactive effects including precipitation, exchange and runoff in the investigated hydrological cycle.

## Figures and Tables

**Figure 1 ijerph-17-00773-f001:**
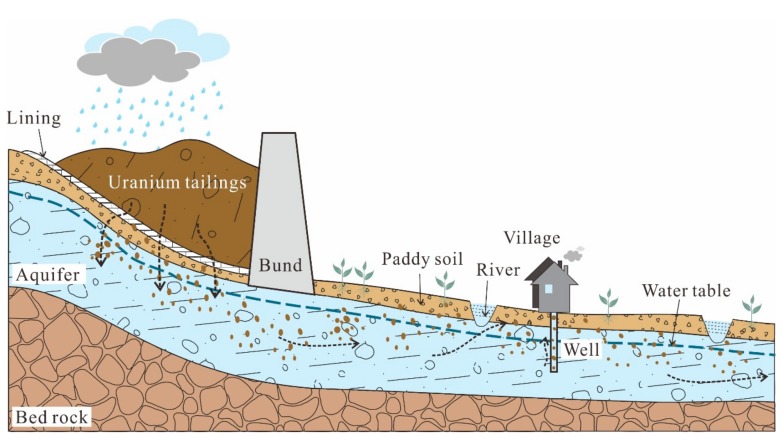
Conceptual model of U cycling around a U mill tailings pond.

**Figure 2 ijerph-17-00773-f002:**
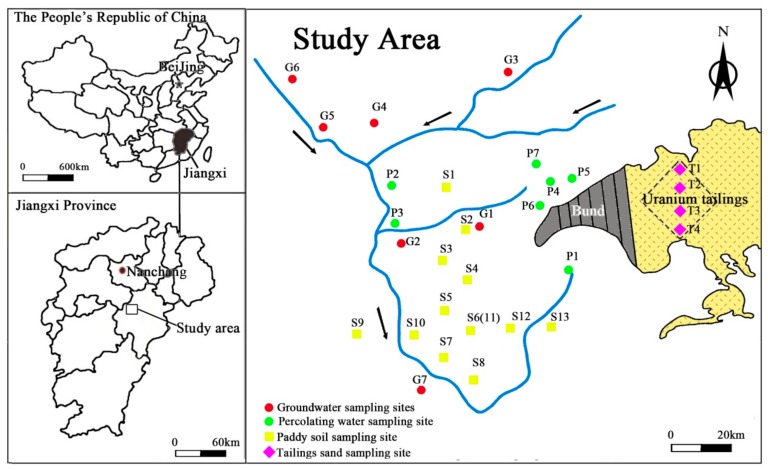
Location of sampling sites.

**Figure 3 ijerph-17-00773-f003:**
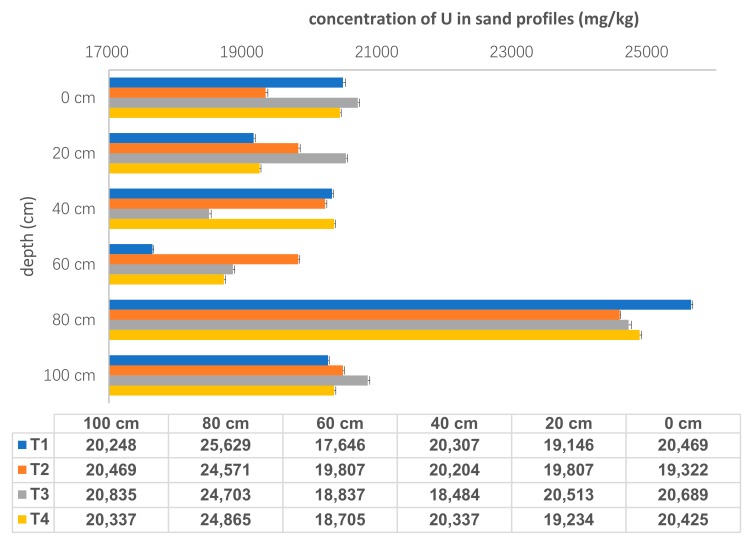
The concentrations of U in sand samples (*n* = 24).

**Figure 4 ijerph-17-00773-f004:**
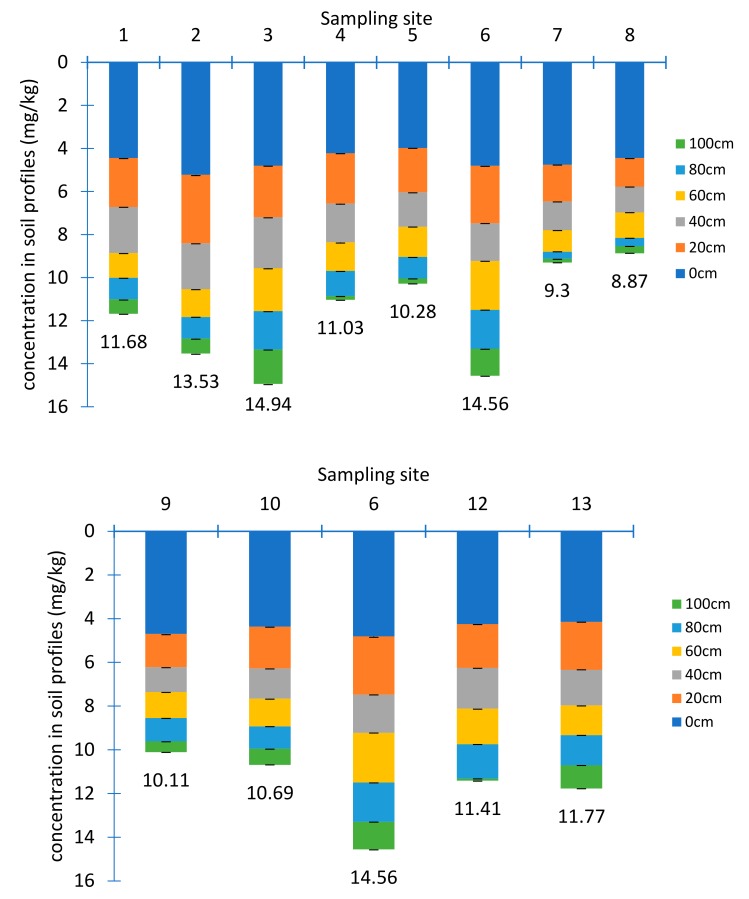
The U concentrations in soil profiles (*n* = 72) (a. longitudinal section, b. transversal section).

**Figure 5 ijerph-17-00773-f005:**
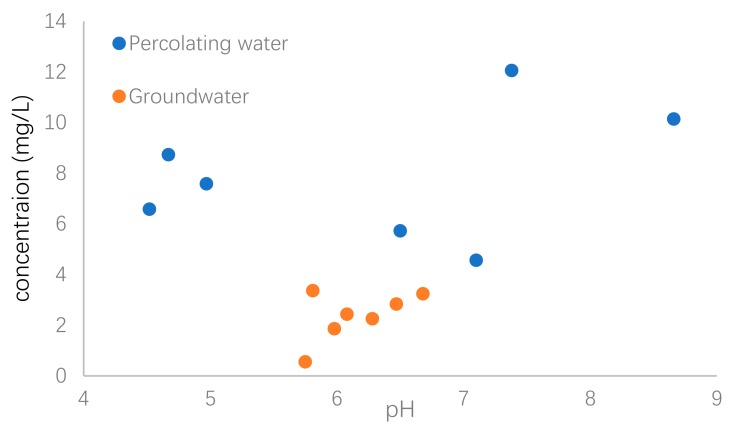
Variation of pH and U concentrations in percolating water and groundwater.

**Table 1 ijerph-17-00773-t001:** U concentrations in percolating water samples (*n* = 7).

Points	Description	pH	Concentration (mg/L)
P1	Tailings pond discharge water	7.38	12.05 ± 0.04
P2	Sewage treatment plant discharge water	7.10	4.56 ± 0.02
P3	A village drain water	6.50	5.72 ± 0.01
P4	B village drain water	4.67	8.73 ± 0.02
P5	Leakage water from top of tailings pond dam	8.66	10.14 ± 0.03
P6	Leakage water from bottom of tailings pond dam	4.97	7.58 ± 0.00
P7	Farmland water near the bottom of tailings pond dam	4.52	6.58 ± 0.01

**Table 2 ijerph-17-00773-t002:** Concentrations of U in groundwater samples (*n* = 7).

Points	Description	pH	Concentration (mg/L)
G1	Drinking well in village A	5.75	0.55 ± 0.04
G2	A new well in village A	5.81	3.36 ± 0.02
G3	A new well in village C	6.68	3.24 ± 0.00
G4	Drinking well in village D	5.98	1.86 ± 0.01
G5	Drinking well in village D	6.08	2.43 ± 0.01
G6	Spring in village E	6.47	2.83 ± 0.02
G7	Spring in village F	6.28	2.25 ± 0.01
Threshold for tailings and inactive sites of USEPA			0.044
Threshold of radiation and environmental protection in U mining of china			0.05

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
