# Peer review of "Occurrence and Distribution of Uranium in a Hydrological Cycle around a Uranium Mill Tailings Pond, Southern China"

_ijerph, 2020, doi:10.3390/ijerph17030773_

Round 1
Reviewer 1 Report
I read the manuscript interestingly, and the authors have worked on very important question. However, there are still some issues that have to be addressed by the authors before considering the manuscript for publication. My comments are detailed below.
Specific comments:
In the “Abstract”, and throughout the manuscript, authors should review sentences such as “...few works have been reported considering U levels in a whole ecosystem.” or “Results showed that U concentrations have different levels in ecosystems with a range of...”– This study focuses only on abiotic components, no biology is invoked here, so the authors cannot speak of “ecosystems”.
In the “Abstract”, and throughout the manuscript, authors should review the use of the term “risk”. The “potential ecological hazard index” and “Geo-accumulation index” are of interest, but chemistry is generally only part of the triad used to assess risk in soil or sediment. No biology is invoked here. Chemical levels are not risk assessments – they are hazard assessments – risk is a function of hazard and exposure. While looking at total concentration elements is part of that overall assessment, the authors do not really engage in what their data mean in terms of the concepts of risk (and risk management). Throughout the manuscript, authors should review the use of the term “risk”.
Keywords
Authors should rephrase keywords. Do not use words or terms in the title as keywords: the function of keywords is to supplement the information given in the title. Words in the title are automatically included in indexes, and keywords serve as additional pointers.
Introduction
Lines 42-43: Association of cancer and uranium ingestion is yet to be established. While writing the health context authors should quote reference from agencies such as UNSCEAR, ICRP, BEAR, NCRP, WHO.
Materials and methods
In this section you should specify the characteristics of all equipments (report model, brand name, city and country of manufacturer).
Please include the month and year of the sampling.
About sampling and soil sample treatment, this sub-section needs more information. For example: The samples were dried? How long were the samples dried for? At what temperature?
This section also should give more details about Quality Assurance and Quality Control. Authors should indicate the obtained accuracy values. Were used reference materials (soil and water)? If so, these should be listed. If so, these should be listed. It would be also interesting to provide the reader with limits of detection/determination of analyzed element.
In addition, the “Materials and methods” section needs a subsection on statistical tests. The statistical analysis of data is not described in the methods, and there are no comparisons for significant differences. I think this should be improved.
In general, the authors should be more precise in describing the experimental methods, and the statistical analyses.
Results and discussion
Tables and Figures – Authors should indicate the number of samples (n =).
Line 104: The authors refer to “Total concentrations of U”. However, they used a digestion method with only sulfuric acid. Therefore, no hydrofluoric acid was used, so the silicates were not dissolved and therefore the measured concentrations are not total. They are only pseudo-totals. Since this method is not intended to accomplish total decomposition of the sample, the extracted analyte concentrations may not reflect the total content in the sample. Therefore, if the authors intended to obtain the total concentrations, the samples digestion method was poorly chosen.
Authors have serious concept problems; which are observed in relation to the “content” and “level” vs concentration.
Lines 190-191: The authors refer “a good positive correlation”. However, they make no reference to the method used to calculate the correlation coefficients (in the “Materials and Methods” section.) And where are the correlation data? As a matter of page limit, the authors can present these results as supplementary material.
Conclusions
The conclusion is quite long and it contains also discussion parts. Authors should avoid repeating the results of the study, or present an abstract. The “Conclusions” section has a different meaning from the “Abstract” section. Authors should present here the main findings, including that which can contribute to new knowledge. Authors can also suggest possible implications and applications of this knowledge and “pathways” for future work.
Author Response
Response to Reviewer 1 Comments
Dear Reviewer,
Thank you very much for the hard work and kindness help toward our manuscript ijerph-671946. We have read the recommendations carefully and concluded that all advice and comments the reviewers had proposed to our paper are useful to make it reasonable and more scientific. In the revised version, we have modified the inappropriate content based on the recommendations. We did try our best to give the reviewers a satisfied answer. And we have summarized our revisions toward the comments as below. Please have a consideration of them:
Point 1: In the “Abstract”, and throughout the manuscript, authors should review sentences such as “...few works have been reported considering U levels in a whole ecosystem.” or “Results showed that U concentrations have different levels in ecosystems with a range of...”– This study focuses only on abiotic components, no biology is invoked here, so the authors cannot speak of “ecosystems”.
Response 1: We much appreciate the reviewers for all the remarks, which would substantially improve the manuscript. We have carefully checked the abstract and revisions have been made accordingly (Line:13, 17, 22, 39). The term “ecosystem” was change to “soil-water system” in the text (Line: 13).
Point 2: In the “Abstract”, and throughout the manuscript, authors should review the use of the term “risk”. The “potential ecological hazard index” and “Geo-accumulation index” are of interest, but chemistry is generally only part of the triad used to assess risk in soil or sediment. No biology is invoked here. Chemical levels are not risk assessments – they are hazard assessments – risk is a function of hazard and exposure. While looking at total concentration elements is part of that overall assessment, the authors do not really engage in what their data mean in terms of the concepts of risk (and risk management). Throughout the manuscript, authors should review the use of the term “risk”.
Response 2: We agree with the reviewer. We have made corresponding revisions according to the comments (Line:12, 22, 159, 226, 231).
Point 3: In the “Keyword”, Authors should rephrase keywords. Do not use words or terms in the title as keywords: the function of keywords is to supplement the information given in the title. Words in the title are automatically included in indexes, and keywords serve as additional pointers..
Response 3: We much thank the reviewer for the comments. The keywords were rephrased as the reviewer suggested (Line:27-28).
Point 4: In the “ Introduction”, Lines 42-43: Association of cancer and uranium ingestion is yet to be established. While writing the health context authors should quote reference from agencies such as UNSCEAR, ICRP, BEAR, NCRP, WHO.
Response 4: We agree with the reviewer’s comments. A related reference from UNSCEAR was added for descriptions of the health effect of U (see below) (Line:42- 44, 323-326).
UNSCEAR. Sources. Effects and risks of ionizing radiation. Volume I: Sources: Report to the general assembly, with scientific annexes. UNSCEAR 2016 Report. United Nations Scientific Committee on the Effects of Atomic Radiation. United Nations publication sales No. E.17.IX.1. United Nations, New York, 2017.
Point 5: In the “Materials and methods”, In this section you should specify the characteristics of all equipments (report model, brand name, city and country of manufacturer).Please include the month and year of the sampling.
Response 5: We much thank the reviewer for the critical comments. The “Materials and methods” part was rewritten (See Materials and methods) and information on the equipment used for analysis has been detailed (Line: 105-107).
Point 6: In the “Materials and methods”, About sampling and soil sample treatment, this sub-section needs more information. For example: The samples were dried? How long were the samples dried for? At what temperature?
Response 6: We agree with the comments given the reviewer. We rewrote the “Materials and methods” part and detailed information about sampling and pretreatment for each type of samples was added in the text (Line:107-114).
Point 7: In the “Materials and methods”, This section also should give more details about Quality Assurance and Quality Control. Authors should indicate the obtained accuracy values. Were used reference materials (soil and water)? If so, these should be listed.It would be also interesting to provide the reader with limits of detection/determination of analyzed element.
Response 7: We agree with the reviewer. During the course of the experiment, 6 standard solutions and 1 blank were used to check the stability of the system for U measurements. Every solution was ran 3 times, and the mean measured values were used to establish the standard curve. Each sample were tested 3 times as well, and then the average value was obtained based on the standard curve. For every sample, the standard errors (SD) of the measured values were less than 1 %, and the precision was +/- 0.01 mg/L. Related descriptions on specifying the quality assurance and quality control were added in the section of “Materials and methods” (Line: 115-119).
Point 8: In addition, the “Materials and methods” section needs a subsection on statistical tests. The statistical analysis of data is not described in the methods, and there are no comparisons for significant differences. I think this should be improved.
Response 8: According to the reviewer’s comments, we have added a subsection about the statistical tests (performed using SPSS 20.0) in the section of “Materials and methods” for comparisons of data differences (Line:119-122).
Point 9: In the “Materials and methods”, In general, the authors should be more precise in describing the experimental methods, and the statistical analyses.
Response 9: We much thank the reviewer’s useful recommendation. We added a detailed description about the experimental methods and statistical analyses in the section of “Materials and methods” (Line:107-119).
Point 10: In the “Results and discussion”, Tables and Figures – Authors should indicate the number of samples (n =).
Response 10: Done (Line:138, 163, 197, 243).
Point 11: In the “Results and discussion”, Line 104: The authors refer to “Total concentrations of U”. However, they used a digestion method with only sulfuric acid. Therefore, no hydrofluoric acid was used, so the silicates were not dissolved and therefore the measured concentrations are not total. They are only pseudo-totals. Since this method is not intended to accomplish total decomposition of the sample, the extracted analyte concentrations may not reflect the total content in the sample. Therefore, if the authors intended to obtain the total concentrations, the samples digestion method was poorly chosen.
Response 11: We must thank the reviewer for the careful observation and we are sorry for not presenting a detailed description of the analytical methods. Indeed, it was our mistake to not state the preprocess correctly. Actually, we used sulfuric acid to deal with the tailing sand samples, and used hydrofluoric acid and nitric acid to deal with paddy soil samples. Related descriptions have been added in the text (Line: 112-114).
Point 12: In the “Results and discussion”, Authors have serious concept problems; which are observed in relation to the “content” and “level” vs concentration.
Response 12: According to the reviewer’s comments. We have checked the use of these terms pointed out by the reviewer and changes have been made in the revised manuscript (Line: 16, 23, 53, 163, 171, 192, 193, 197, 204, 206, 243, 245, 249, 257, 262, 268, 270, 275, 280, 282, 285, 289, 291).
Point 13: In the“Results and discussion”, Lines 190-191: The authors refer “a good positive correlation”. However, they make no reference to the method used to calculate the correlation coefficients (in the “Materials and Methods” section.) And where are the correlation data? As a matter of page limit, the authors can present these results as supplementary material.
Response 13: We agree with the reviewer’s comments. The correlation coefficients were determined using SPSS 20.0. Related revisions have been made in the “Materials and Methods” section (Line:119-122). The correlation coefficient was added as suggested by the reviewer (Line: 221).
Point 14: In the “Conclusions”, The conclusion is quite long and it contains also discussion parts. Authors should avoid repeating the results of the study, or present an abstract. The “Conclusions” section has a different meaning from the “Abstract” section. Authors should present here the main findings, including that which can contribute to new knowledge. Authors can also suggest possible implications and applications of this knowledge and “pathways” for future work.
Response 14: We much appreciate the reviewer’s remarks. The conclusion was shortened and refined. Revisions were made according to the comments (Line:277-294).
Thank you very much for your kindness recommendation!
It is our pleasure to answer your questions! These professional comments are very useful to the revised version of this paper and they are also important to our further research! Again, thank you for your kindness and carefulness for our paper! We wish the revised version can give you a complete satisfaction.
Bai Gao
Dec. 18, 2019

Reviewer 2 Report
The manuscript paper is interesting from scientifical point of view, but before publication in International Journal of Environmental Research and Public Health should be corrected and improved on:
The subsection 2. “Samples collection and analysis?” should be improved on important analytical validation parameters, like: accuracy, precision and blanks during determination of U in analyzed samples by ICP-OES technique. All results of U concentration in analyzed samples given in tables 1 and 2 as well as in text of manuscript or figure 3 and 4 should be given with measurement uncertainty (or standard deviation). The all results of U concentration content in the manuscript paper should be given with 3 significant numerals (e.g., instead 123.45 should be 123+/- measurement error).Author Response
Response to Reviewer 2 Comments
Dear Reviewers,
Thank you very much for your hard work and kindness help toward our manuscript ijerph-671946. We have read your recommendations carefully and concluded that all advices and comments you had proposed to our paper are useful to make it reasonable and more scientific. In the revised version, we have modified the inappropriate content based on your recommendations. We did try our best to give you a satisfied answer. And we have summarized our revisions toward your comments as below. Please have a consideration of them:
Point 1: Samples collection and analysis?”should be improved on important analytical validation parameters, like: accuracy, precision and blanks during determination of U in analyzed samples by ICP-OES technique.
Response 1: We must thank the reviewers for all the comments, which would greatly improve the quality of our manuscript. According to the comments, we rewrote the Materials and methods part where information on analytical methods and validation has been detailed (Line: 86-122).
Point 2: All results of U concentration in analyzed samples given in tables 1 and 2 as well as in text of manuscript or figure 3 and 4 should be given with measurement uncertainty (or standard deviation).
Response 2: We agree with the reviewer’s comments. Standard deviations for the determined data have been added in the section of Material and method (Line:118-122) .
Point 3: The all results of U concentration content in the manuscript paper should be given with 3 significant numerals (e.g., instead 123.45 should be 123+/- measurement error).
Response 3: We much thank the reviewer for the comments. Indeed, the reviewer suggested a good way to show data. Actually, the present U concentrations are average values, and U detect limitation was determined to be 0.01 mg/L.This is sufficient enough to obtain an accurate analytical result due to high aqueous U concentrations. Detailed information on analysis has been added in the text (Line:118-119).
Thank you very much for your kindness recommendation!
It is our pleasure to answer your questions! These professional comments are very useful to the revised version of this paper and they are also important to our further research! Again, thank you for your kindness and carefulness for our paper! We wish the revised version can give you a complete satisfaction. Thank you again for your useful comments on our paper!
Bai Gao
Dec. 18, 2019

Reviewer 3 Report
The paper "Occurrence and distribution of uranium in a hydrological cycle around a uranium mill tailings pond, southern China" describes the occurrence and distribution of uranium in the vicinity of mill tailings pond. In my opinion, the paper requires a major revision prior to publication.
For starters, the title does not match the rest of the text. English is very bad; grammar, spelling, etc. The Method part is written very confusingly and it is not clear what was sampled, for what purpose, what analyzes were performed and on what samples. The results also need to be revised. Table one shows 4 samples and Table 2 seven samples, and a total of 110 samples were analyzed ????
The authors provide conclusions that are not results-based. The discussion should not only be refined and expanded, but also structured differently. I suggest that the authors look at similar papers in the field and make the extra effort to more clearly present their results, give adequate discussion, and revise their conclusions accordingly.
The topic is worth exploring, and I believe the results obtained are also worth publishing, but not in this form.
Author Response
Response to Reviewer 3 Comments
Dear Reviewers,
Thank you very much for your hard work and kindness help toward our manuscript ijerph-671946. We have read your recommendations carefully and concluded that all advices and comments you had proposed to our paper are useful to make it reasonable and more scientific. In the revised version, we have modified the inappropriate content based on your recommendations. We did try our best to give you a satisfied answer. And we have summarized our revisions toward your comments as below. Please have a consideration of them:
Point 1: For starters, the title does not match the rest of the text. English is very bad; grammar, spelling, etc.
Response 1: We must thank the reviewer for the careful observation and we are sorry for some spelling and grammar errors that had made troubles for a better understanding of our work. The manuscript has been examined carefully for times and language errors have been revised according to the comments (Line: 11, 13-25, 33-34, 42-44, 49, 53-54, 58-61, 63, 65, 71, 73, 75, 80, 83, 128-135, 145, 159, 171-173, 178, 184-186, 189-190, 193-194, 200-204, 210-216, 223-232, 235-240, 245, 249-250, 252, 259-263,267-275).
Point 2: The Method part is written very confusingly and it is not clear what was sampled, for what purpose, what analyzes were performed and on what samples.The results also need to be revised.
Response 2: We agree with the comments given by the reviewer. According to the comments, we rewrote the Materials and Methods part where information on purposes of sampling, processing and analysis for each type of samples has been detailed (Line: 86-122).
Point 3: Table one shows 4 samples and Table 2 seven samples, and a total of 110 samples were analyzed ????
Response 3: We much thank the reviewer for the critical comments. We have collected 4 types of samples, including tailing sand, percolating water, paddy soil and groundwater. For tailing sand, samples were collected from 6 layers (0 cm, 20 cm, 40 cm, 60 cm, 80 cm and 100 cm) at 4 sampling points, so the total tailing sand sample were 24 (4 points * 6 samples/point). Similarly, paddy soils were sampled from 6 layers at 12 sampling points, which means that 72 paddy soil samples were collected (12 points * 6 samples/point). Plus 7 percolating water samples and 7 groundwater samples, a total of 110 samples were obtained. We have added a detailed description to specify the amount of samples collected in the section of “Materials and methods” and indicated the number of samples (n=) in each table and figure (Line:138, 163, 197, 243).
Point 4: The authors provide conclusions that are not results-based. The discussion should not only be refined and expanded, but also structured differently. I suggest that the authors look at similar papers in the field and make the extra effort to more clearly present their results, give adequate discussion, and revise their conclusions accordingly.
Response 4: We agree with the reviewer’s comments and much appreciate the suggestions. The conclusion part has been rewritten, and the manuscript has been revised for times according to the comments proposed by the reviewer (Line:277-294).
Thank you very much for your kindness recommendation!
It is our pleasure to answer your questions! These professional comments are very useful to the revised version of this paper and they are also important to our further research! Again, thank you for your kindness and carefulness for our paper! We wish the revised version can give you a complete satisfaction.
Bai Gao
Dec. 18, 2019

Reviewer 4 Report
The authors have made an assessment to the occurrence and distribution of U in a soil-water system surrounding a uranium mill tailings pond. Overall, the study is interesting and may report valuable findings. Yet, there are a few issues that must be addressed before proceeding. More specifically:
Language: The manuscript would benefit from a revision from a native English speaker or professional editing services. The vocabulary and grammar need to be polished and the verb tenses need to be rechecked.
Materials and methods: More details about the mine, including its location (with coordinates) would be valuable.
Materials and methods: In the response to reviewer document, please clarify if only 1 replicate was collected per sampling depth for all types of samples (tailings, soil, percolating and groundwater)
Page 4, Line 104: "As shown in Fig. 3"
Figure 3: What do T1-T4 mean? (not explained in the figure legend). Authors are advised to include in-text description of G1-G7, S1-S13, P1-P7, and T1-T4 on section 2.2, so that the reader understands what these codes correspond to.
Author Response
Response to Reviewer 4 Comments
Dear Reviewers,
Thank you very much for your hard work and kindness help toward our manuscript ijerph-671946. We have read your recommendations carefully and concluded that all advices and comments you had proposed to our paper are useful to make it reasonable and more scientific. In the revised version, we have modified the inappropriate content based on your recommendations. We did try our best to give you a satisfied answer. And we have summarized our revisions toward your comments as below. Please have a consideration of them:
Point 1: Language: The manuscript would benefit from a revision from a native English speaker or professional editing services. The vocabulary and grammar need to be polished and the verb tenses need to be rechecked.
Response 1: We must thank the reviewer for the careful observation and we are sorry for some vocabulary and grammar errors that had made troubles for a better understanding of our work. The manuscript has been examined carefully for times and errors of the language used have been revised. Revised portion are marked in red in the Revised Manuscript (Line: 11, 13-25, 33-34, 42-44, 49, 53-54, 58-61, 63, 65, 71, 73, 75, 80, 83, 128-135, 145, 159, 171-173, 178, 184-186, 189-190, 193-194, 200-204, 210-216, 223-232, 235-240, 245, 249-250, 252, 259-263,267-275).
Point 2: Materials and methods: More details about the mine, including its location (with coordinates) would be valuable.
Response 2: Thanks for the reviewer’s suggestions. Due to the agreement with the project management agency, the location of the research area is confidential, so we are sorry that we are unable to provide more details about the mine, and we are eager to get the reviewer’s understanding.
Point 3: Materials and methods: In the response to reviewer document, please clarify if only 1 replicate was collected per sampling depth for all types of samples (tailings, soil, percolating and groundwater)
Response 3: We agree with the reviewer. For tailings, samples were collected from 6 layers (0 cm, 20 cm, 40 cm, 60 cm, 80 cm and 100 cm) at 4 sampling points (4 points * 6 samples/point). Similarly, paddy soils were sampled from 6 layers at 12 sampling points (12 points * 6 samples/point). Percolating water samples and groundwater samples were taken from different locations rather than done replicate. We have added a description to specify the amount of samples collected and sampling methods in the “Materials and methods” section (Line:86-122).
Point 4: Page 4, Line 104: "As shown in Fig. 3"
Response 4: Corrections were made according to the comment (Line:128).
Point 5: Figure 3: What do T1-T4 mean? (not explained in the figure legend). Authors are advised to include in-text description of G1-G7, S1-S13, P1-P7, and T1-T4 on section 2.2, so that the reader understands what these codes correspond to.
Response 5: We agree with the reviewer’s comments. We have added specific explanation for the codes used in the text (Line:90-98).
Thank you very much for your kindness recommendation!
It is our pleasure to answer your questions! These professional comments are very useful to the revised version of this paper and they are also important to our further research! Again, thank you for your kindness and carefulness for our paper! We wish the revised version can give you a complete satisfaction. Thank you again for your useful comments on our paper!
Bai Gao
Dec. 18, 2019

Round 2
Reviewer 1 Report
This is the second version of this manuscript, and it improves in both readability and scientific clarity with revision, so I commend the authors for their perseverance. I have no additional comments.
Author Response
Dear Reviewers,
Thank you very much for your kindness help toward our manuscript ijerph-671946.
Point 1: This is the second version of this manuscript, and it improves in both readability and scientific clarity with revision, so I commend the authors for their perseverance. I have no additional comments.
Response:We sincerely thank the reviewer for your patient comments and suggestions. The review process of this manuscript is a very valuable learning experience for me. In the future scientific research, I will uphold a rigorous scientific attitude and make continuous efforts. Finally, thank you for your approval of our article.
Reviewer 4 Report
Although a few aspects of the manuscript were improved, I still have serious concerns regarding the significance of the results. Hence, I urge the authors to clarify the following points:
Page 3, line 119: "Finally, data statistical analysis was performed by SPSS 20.0 to detect the correlation values between the content of organic matter and the concentration of U." - how did you assess this correlation? which method?
Page 3, line 121: "Significant differences among different sampling were considered at p < 0.05." - Needs clarification. Which samples were compared for significant difference? Which statistical method was used for the comparison? Where are those results?
Author Response
Dear Reviewers,
Thank you very much for your careful observation and kindness help toward our manuscript ijerph-671946. The comments you had proposed are useful to improve the rigor of our paper. We did try our best to summarized our revisions toward your comments as below. Please have a consideration of them:
Point 1: Page 3, line 119: "Finally, data statistical analysis was performed by SPSS 20.0 to detect the correlation values between the content of organic matter and the concentration of U." - how did you assess this correlation? which method?
Response: We are much grateful to the reviewer for the constructive comments, which greatly improve the scientific quality of our manuscript. According to the comments given by the reviewer, a detailed information on how to analyze the correlation between U concentrations and organic matter concentrations using Person Correlation Analysis method of SPSS 20.0 was presented in the text (line:119-121, 221-223). The imported data were shown below (Table 1).
Table 1 Average concentration of U and Organic matter
| Layer below the surface | U concentration/ mg·L-1 | Organic matter concentration/g·kg-1 |
| 0 cm | 4.52 | 43.3 |
| 20 cm | 2.15 | 32.4 |
| 40 cm | 1.67 | 23.7 |
| 60 cm | 1.50 | 16.4 |
| 80 cm | 1.21 | 10.3 |
| 100 cm | 0.69 | 7.7 |
Point 2: Page 3, line 121: "Significant differences among different sampling were considered at p < 0.05." - Needs clarification. Which samples were compared for significant difference? Which statistical method was used for the comparison? Where are those results?.
Response: We agree with the reviewer. In order to ensure the quality assurance of the data shown in the manuscript, the concentrations of U in all samples, including tailings sand, paddy soil, percolating water and groundwater were tested for significance differences using the K-S inspection method (please see the attachment ). The data were used for discussion based on their distribution which follows the normal distribution at a 0.05 confidence interval. According to the reviewer’s comments, related descriptions have been added in the text (line: 121-125).
Thank you very much for your kindness recommendation!
These professional comments are very useful to the revised version of this manuscript. We wish the revised version can give you a complete satisfaction.
Bai Gao
Dec. 27, 2019

Round 3
Reviewer 4 Report
The authors' multiple clarifications (and changes) on their statistical analysis, have failed to convince me about the significance of their results and the reliability of the methods employed. Note that the first review pointed out the absence of replicates (1 sample per sampling depth); then the second version of the manuscript mentioned a mysterious ANOVA comparison (without replicates?!); now there is no ANOVA but a simple K-S distribution test. Moreover, the Pearson correlation is presented as an R squared (R2) coefficient.
Author Response
Dear Reviewers, First of all, we would like to express our sincere apologies for not being able to give you a satisfactory reply toward the comments you have gave to our manuscript ijerph-671946. We explained in detail the section of statistical analysis problems that made you confused even doubt as below. Please have a consideration of them: Point 1: The authors' multiple clarifications (and changes) on their statistical analysis, have failed to convince me about the significance of their results and the reliability of the methods employed. Note that the first review pointed out the absence of replicates (1 sample per sampling depth); then the second version of the manuscript mentioned a mysterious ANOVA comparison (without replicates?!); now there is no ANOVA but a simple K-S distribution test. Moreover, the Pearson correlation is presented as an R squared (R2) coefficient. Response: We much thank the reviewer for the critical comments. The complete data acquisition and analysis processes are as follows: Firstly, the tailings sand samples at 4 points were collected with depth, each depth we only collect 1 kg with no replicates. This was confirmed in the first round review, as pointed out by the reviewer. The same method was used to collect paddy soil samples from 12 points. Secondly, all of the solid samples (tailings sand and paddy soil) were analyzed after full digestion, and then were transferred from liquid concentration (mg/L) to solid concentration (mg/kg) based on the obtained results. Not only standard solution but also each sample was tested 3 times. The relative error for the standard curve was better than 1%, and the mean value of the 3 times was used as the result of each sample for subsequent discussion. Comparisons for U concentration were performed between the repeated analyses rather than replicated samples. Finally, a method of K-S inspection [1] was employed for each type sampling data to check the reliability of sample collection. More detailed information on the section of analysis has been presented in the text according to the comments (Line:117-128, 226-229). [1] Chen Jingying. Pollution characteristics and migration-transformation mechanism of uranium in soil around uranium tailings pond[D]. Nanchang University, 2019. Thank you very much for your critical recommendation! These professional comments are very important and will make our research more rigorous! We wish you a happy New Year! Bai Gao Jan. 04, 2020